# ECG-Based Semi-Supervised Anomaly Detection for Early Detection and Monitoring of Epileptic Seizures

**DOI:** 10.3390/ijerph20065000

**Published:** 2023-03-12

**Authors:** Apostolos Karasmanoglou, Marios Antonakakis, Michalis Zervakis

**Affiliations:** Digital Image and Signal Processing (DISPLAY) Laboratory, School of Electrical and Computer Engineering, Technical University of Crete (TUC), Akrotiri Campus, 73100 Chania, Greece; mantonakakis@tuc.gr (M.A.); mzervakis@tuc.gr (M.Z.)

**Keywords:** epilepsy, seizure prediction, electrocardiogram, semi-supervised, anomaly detection, heart rate variability

## Abstract

Epilepsy is one of the most common brain diseases, characterized by frequent recurrent seizures or “ictal” states. A patient experiences uncontrollable muscular contractions, inducing loss of mobility and balance, which may result in injury or even death during these ictal states. Extensive investigation is vital to establish a systematic approach for predicting and informing patients about oncoming seizures ahead of time. Most methodologies developed are focused on the detection of abnormalities using mostly electroencephalogram (EEG) recordings. In this regard, research has indicated that certain pre-ictal alterations in the Autonomic Nervous System (ANS) can be detected in patient electrocardiogram (ECG) signals. The latter could potentially provide the basis for a robust seizure prediction approach. The recently proposed ECG-based seizure warning systems utilize machine learning models to classify a patient’s condition. Such approaches require the incorporation of large, diverse, and thoroughly annotated ECG datasets, limiting their application potential. In this work, we investigate anomaly detection models in a patient-specific context with low supervision requirements. Specifically, we consider One-Class SVM (OCSVM), Minimum Covariance Determinant (MCD) Estimator, and Local Outlier Factor (LOF) models to quantify the novelty or abnormality of pre-ictal short-term (2–3 min) Heart Rate Variability (HRV) features of patients, trained on a reference interval considered to contain stable heart rate as the only form of supervision. Our models are evaluated against labels that were either hand-picked or automatically generated (weak labels) by a two-phase clustering procedure for samples of the “Post-Ictal Heart Rate Oscillations in Partial Epilepsy” (PIHROPE) dataset recorded by the Beth Israel Deaconess Medical Center, Harvard Medical School, Boston, Massachusetts, achieving detection in 9 out of 10 cases, with average AUCs of over 93% across all models and warning times ranging from 6 to 30 min prior to seizure. The proposed anomaly detection and monitoring approach can potentially pave the way for early detection and warning of seizure incidents based on body sensor inputs.

## 1. Introduction

In 2022, over 50 million people were reported to suffer from epilepsy worldwide [1] making it the fourth-most-common brain disease after migraine, stroke, and Alzheimer’s [2]. Patients with epilepsy suffer from frequent recurrent seizures after bursts of electrical activity in the brain, which manifest in uncontrollable muscle contractions, loss of consciousness, loss of muscle control and balance, and disruption to the sensory system. The onset of these episodes may not be perceived ahead of time, so they may be physically impairing and dangerous. In a recent study [3] conducted on 72 adults, 55 of them (76.3%) reported that they had suffered seizure-related injuries throughout their lifespan, while 17 (23.6%) suffered injuries within the year that the study took place. The study also found that sufferers of epilepsy are less likely to partake in risky day-to-day activities (driving, cooking, ironing, etc.) and reported on potential strain on their quality of life. More severely, it was estimated that in symptomatic epilepsy (epilepsy induced as a symptom of brain injury), female sufferers lose up to 11 years of life and males up to 13 years [4], while another study reports similar findings of 10–11 years of lost life for females and 11–12 for males [5]. Notably, fatalities of unknown cause during seizures or “Sudden Unexpected Death in Epilepsy” (SUDEP) [6] occur in approximately 0.9% of intractable epilepsy cases [7]. Definitive causes for patient mortality remain unknown; however, they have been linked to respiratory and cardiac dysfunction [8,9,10,11,12].

In recent years, certain machine learning techniques have been shown to provide the means for monitoring patient ictal state and the detection of seizures ahead of time. Interest has also been raised in incorporating these methods into smart wearable devices for personal and clinical use [13]. These techniques detect abnormal physiological activities and prompt medical intervention to prevent and alleviate the undesired consequences of this condition. Electroencephalogram (EEG) signals are typically utilized for this purpose. In terms of single neuron dynamics, epileptic seizures have been found to follow heterogeneous neuronal spiking activity minutes prior to their onset [14], the effect of which is detectable in EEGs.

Recent studies have demonstrated the potential of electrocardiogram (ECG) signals as an alternative or complementary modality for this purpose [14,15,16,17]. This is possible as the alterations in the Autonomic Nervous System (ANS) occur prior to an ictal state, resulting in modified cardiac behavior, and are detectable by monitoring the short-term Heart Rate Variability (HRV) parameters of a patient. The term “short-term HRV” refers to HRV parameters estimated over a short time window, which have a different medical meaning from long-term HRV parameters [18,19,20]. In this study, we make use of short-term HRV. Short-term HRV parameters can be monitored by relatively low-cost wearable devices for patient surveillance and data-informed medical counseling [13,21,22,23]. Monitoring heart rate may be especially useful outside the context of forecasting seizure onset, considering the condition’s link to cardiac function disruption and the latter’s connection to SUDEP cases.

Several approaches to identifying pre-ictal physiological alterations with the use of machine learning algorithms have been developed [15,24,25,26,27,28], aiming to discriminate between pre-ictal and inter-ictal cardiac states based on their statistical footprints. Discriminative models based on statistical properties of the RR Interval (RRI) series of an ECG signal act as the base for most of the previously mentioned studies. An RR interval is the time interval between successive QRS complex R peaks present in the ECG, which correspond to heart beats. Machine learning approaches based on HRV usually compute a set of short-term (2–3 min) statistical features of the RR series and use them as learning data to train a machine learning algorithm, such as Support Vector Machines (SVM), Decision Trees, K-Nearest Neighbors (K-NN), Kernel Spectral Clustering, etc.

Despite the significance of the work in demonstrating the statistically distinguishable nature of inter-ictal and pre-ictal segments, it is limited in addressing some practical matters regarding its deployment. More specifically, major concerns with these approaches are listed as follows:(1)They require enrollment: Patient-specific deployment of discriminative model methods presupposes the existence of prior recordings of a patient’s seizures, as fitting classifier models requires samples from both inter-ictal and pre-ictal classes. This means that when designing a patient-specific system, a patient would have to enroll by undergoing a strenuous ECG recording session, in order to experience one or several seizures to personalize the device with specific pre-ictal heart rate parameters.(2)They are data hungry: Patient-agnostic deployment, on the other hand, would be dependent on the existence of large, difficult-to-obtain datasets. To train robust classifiers with the ability to generalize across different patient and seizure cases, a large amount of seizure recordings would need to be acquired and reliably labeled. This, in turn, gives rise to the necessity of conducting lengthy data acquisition sessions that require ethical clearance.(3)They are trained with unreliable labels: ANS disturbances occur at different time intervals for each epileptic seizure, making the procedure of reliably labeling ECG segments subjective to the human ability to distinguish the patterns of these disturbances.(4)They are potentially error-prone when presented with novel samples: There is no guarantee that novel ANS disturbances not included in a dataset will be detected using these methods. It is possible that in a two-class classifier that discriminates ECG segments as inter-ictal or pre-ictal, certain ECG abnormalities remain within the boundary of the wrong class when mapped onto the feature space. However, an efficient prediction system that is sensitive to all deviations from a pre-defined normal HRV pattern is not compromised in this scenario.

Our proposed solution to these problems is to substitute the classifiers with anomaly detection schemes capable of quantifying the “novelty” of a segment of the RR series. Our assumption is that the underlying distribution of HRV patterns during inter-ictal and pre-ictal periods diverge significantly, so that the latter patterns are detected as outliers with respect to the distribution of the former. In this sense, anomalies can be detected without requiring prior knowledge of pre-ictal HRV patterns. These models can be deployed in a patient-specific manner, without training on large datasets, but simply with a small reference interval of inter-ictal HRV data, making them highly flexible to each patient case. The setting of a stable HRV reference interval is the only form of supervision these models receive, which is potentially more reliable than using class labels as the patient or a clinician can easily verify their condition as stable. Furthermore, these systems are fundamentally designed to discriminate samples based on their novelty, so they may generalize better in detecting a wide range of HRV disturbances. These features would allow the implementation of a “plug-and-play” seizure detection service that can be calibrated on a specific patient during a non-ictal phase and, subsequently, be used to monitor the patient and detect any heart rate abnormalities that may occur in the future.

The framework we propose is based on unsupervised HRV seizure detection. Similar work in this direction has been done in EEG-based detection [29,30,31,32,33]. Unsupervised identification of the pre-ictal interval using HRV has been carried out by Leal et al. [28], where they examined different clustering algorithms to identify the pre-ictal interval in recorded seizures. These solutions were designed to characterize altered segments of heart rate signals, which is useful in searching for pre-ictal patterns, and thus informing possible seizure prediction. Using clustering methods, however, assumes that relevant pre-ictal alterations are present in the data being clustered, and prediction of seizure onset relies on the inspection of the estimated cluster properties. Although this work is noteworthy in paving the way for unsupervised seizure detection, these attributes make the proposed framework less suitable for establishing an automatic real-time seizure warning system. Although this work is noteworthy in paving the way for unsupervised seizure detection, these attributes make the proposed framework more suitable to post hoc clinical inspection rather than for establishing a real-time seizure warning system. Another significant work in this direction is that of Fujiwara et al. [27], in which anomalies in HRV features were detected using a pair of test statistics: Q-dissimilarity and Hoteling’s $T^{2}$. This latter approach resembles our proposed design; however, it only produces warnings at the onset of HRV variation, and not for the entire duration of the cardiac abnormality. It also exhibits several false positives, leaving room for improved alternatives. To address these issues, we utilize anomaly detection models which do not rely on post hoc discrimination of short-term HRV features but can detect alterations in the form of outliers as they occur. Furthermore, these models do not simply measure feature variation over time to produce warnings, but instead provide a measure for quantifying their similarity to a referential period of heart rate behavior.

We thus evaluate the performance of three commonly used “shallow” anomaly detection techniques—Local Outlier Factor (LOF) [34], Minimum Covariance Determinant (MCD) estimator [35], and One-Class SVM (OCSVM) [36]—in locating HRV novelties present in segments of an RR series. To our knowledge, this is the first time such models have been evaluated in the context of seizure onset prediction and monitoring from patient ECG.

## 2. Methods and Recordings

### 2.1. Pre-Processing

We apply basic pre-processing steps (baseline correction, offline filtering, and visual inspection to reduce non-cardiac activity) to the raw ECG signal as we are not overly concerned with preserving information from its structure. We are concerned with reliably locating the peaks of the QRS complex of each heart cycle. As such, we aim to remove the major effects of baseline wandering as well as high-frequency artifacts from the signal to avoid detecting false peaks or missing others. We apply a bandpass filter with a low-end frequency of 1 Hz and a high-end frequency of 50 Hz. This makes the R peaks more prominent and minimizes the possibility of false peak detection.

The resulting signal is then used for detecting R peaks and deriving the RR series. For this purpose, we used the corresponding utilities provided by the python library *biosppy* [37].

The RR series may still contain some artifacts that are to be removed. To remove any irrelevant outliers in the RR series, such as skipped beats or equipment malfunctions, we selected a maximum threshold for the amount that an RR interval can overshoot proportionally to the absolute value of the local mean. More specifically, if: $$RR_{j}-RR_{j-1}>\tau \cdot AVG_{L}{\left(RR\right)}_{j},$$
where $NN_{j}$ is the j-th NN interval, τ is the decided percentile threshold usually set in the range (30–70%), and $AVG_{L}$ is the L-sample moving average where we typically set L=15,30 , the $RR_{j}$ is replaced by the local median. We will refer to the resulting artifact-free series of normal interbeat intervals as the “NN series”.

### 2.2. Feature Extraction

For each patient, the temporal HRV parameters are defined using the NN series and producing windowed segments consisting of intervals (2–3 min of heart activity in typical cases) to finally extract short-time statistical features. Each segment corresponds to a slide of the window over the NN series by one NN interval, allowing for a fine-grained tracing of the evolution of the HRV parameters during ictal events. A complete set of these features is presented in Table 1. In selecting which features to use, we consulted the common guidelines regarding HRV features [38] as well as lists of features used in related work. The computation of these parameters was performed with the help of the openly available HRV library for python *pyHRV* [39].

These features can be broadly categorized as time domain, frequency domain, and non-linear features according to the *pyHRV* documentation [43]. We specify that frequency domain features were extracted from the NN series Power Spectral Density (PSD), denoted as $PSD_{NN}$, and its corresponding low-frequency (0.04–0.15 Hz) and high-frequency (0.15–0.4 Hz) band segments are denoted as $PSD_{NN}^{LFB},PSD_{NN}^{HFB}.$ The NN series PSD is estimated using Welch’s method [44].

Following feature extraction, we aim to reduce the dimensionality of the representation by use of five-dimensional PCA decomposition, similar to the work of Fujiwara et al. [27]. In the work of Billeci et al. [26], optimal feature selection was performed by an iterative procedure based on improving *p*-values; however, this approach is not possible in an unsupervised context, unlike PCA. In all cases, the low-dimensionality features were obtained by decomposing the feature vectors into principal components estimated during a specified reference interval. We denote the feature-transformed series as $x\left(t\right)$, indexed by the time t of its final R peak. This representation serves as input in training the anomaly detection models and quantifying the novelty of each segment, assigning to each one a novelty score $s\left(t\right)$. The proposed anomaly detection pipeline is outlined by the diagram presented in Figure 1.

### 2.3. Anomaly Detection

Considering a dataset $X=\left\{x_{1},x_{2},\ldots ,x_{n}\right\},x_{i}\in S,i=1,\ldots ,n,$ where $S\subset R^{F}$ is the space of all possible relevant samples transformed into features (which may be unbounded), and $p\left(x\right)$ is an estimate of the ground truth distribution of S based on X, then a set of anomalies in this dataset can be defined as:$$A=\{x\in S:p\left(x\right)<\tau \},$$
where τ is some predefined threshold. In designing anomaly detection systems, our assumption is that the “cluster hypothesis” holds, which states that there exists some positive τ such that the set S\A is nonempty and small with regards to the Lebesgue measure [45]. The intuitive interpretation of this assumption is that data samples generated from the same physical process tend to be clustered in feature space, whereas anomalies tend to lie further away from the central cluster. We investigate the potential use of anomaly detection models in detecting pre-ictal HRV abnormalities. More specifically, the methods we consider are the sample-density-based Local Outlier Factor (LOF) and two types of discriminative models: Minimum Covariance Determinant (MCD) Gaussian and a One-Class Support Vector Machine. In the following sections, we briefly introduce each of these models.

#### 2.3.1. Minimum Covariance Determinant

The MCD anomaly detector functions under the assumption that samples are distributed according to an elliptically symmetrical unimodal distribution with a ground truth location parameter $\mu \in R^{F}$ and a positive definite scatter matrix $\Sigma \in R^{F\times F}.$ The MCD estimator with a tuning parameter of $\frac{n}{2}\leq h\leq n$ is formally defined as the pair where $\hat{\mu }$ is the mean of h samples and $\hat{\Sigma }$ is the corresponding covariance matrix multiplied by a consistency factor scalar constant $c_{0}$. Adding distance-based weighting to the points of the dataset yields the estimates:$$\hat{\mu }=\frac{\sum_{i=1}^{n}w\left(d_{i}^{2}\right)x_{i}}{\sum_{i=1}^{n}w\left(d_{i}^{2}\right)},$$
$$\hat{\Sigma }=\frac{c_{1}}{n}\overset{n}{\underset{i=1}{\sum }}w\left(d_{i}^{2}\right)\left(x_{i}-\hat{\mu }\right){\left(x_{i}-\hat{\mu }\right)}^{T},$$
where $d_{i}=MD\left(x_{i},\mu \left(X\right),Cov\left(X\right)\right)$, i=1,…,n are the standard Mahalanobis Distances of data points using the sample mean and covariance estimates of the dataset $\mu \left(X\right),Cov\left(X\right)$ and $w:R^{F}\to \left[0,+\infty \right)$ is some weighting function. Using these parameters, one can calculate the “robust distances” of a data point as:$$RD\left(x_{i},\hat{\mu },\hat{\Sigma }\right)=MD\left(x_{i},\hat{\mu },\hat{\Sigma }\right).$$

The elliptic envelope yielded by the robust distance for this estimate is derived from the equation $RD\left(x,\hat{\mu },\hat{\Sigma }\right)=\tau_{RD}$, where $\tau_{RD}>0$ is some pre-defined threshold.

Using robust distances instead of a standard Mahalanobis Distance has the advantage of being uncompromised to the masking effect [46] that classical estimates suffer from, causing them to be altered by the existence of outliers in data. The weighted MCD estimator is said to be a high breakdown value estimator, meaning that the smallest fraction of observations that need to be replaced by arbitrary values to make the estimate useless is the highest possible for both the location and the scatter parameter under the assumption of a continuous distribution of samples [47]. These properties make the MCD ideal for designing an outlier-sensitive warning system. In our case, we will use robust distance as a novelty score for new samples.

#### 2.3.2. One-Class Support Vector Machine

The main function of an OCSVM, as in all one-class classifiers [48], is to estimate the parameters of a minimum-volume enclosing sphere of the normal dataset, i.e., a separating boundary between the set of normal samples and the rest of the relevant space (considered an area of outliers) in parallel to standard two-class classification. The objective is to estimate the separating hyperplane between two or more classes. In accordance with the standard formulation for a kernel SVM [49] using a positive semi-definite kernel function k:S×S→R associated with the feature map $r-\langle \phi_{k}\left(x\right),w\rangle \leq \xi_{i},\xi_{i}\geq 0,i=1,\ldots ,n$ via $k\left(x,y\right)=\langle \phi_{k}\left(x\right),\phi_{k}\left(y\right)\rangle ,$ where $F_{k}$ is the related reproducing kernel Hilbert space, we can define the following objective for the OCSVM: $$\underset{\left(w,r,\xi \right)}{min}\left\{{\left|\left|w\right|\right|}^{2}-r+\frac{1}{nv}\overset{n}{\underset{i=1}{\sum }}\xi_{i}\right\},$$
$$s.t\;:\;r-\langle \phi_{k}\left(x\right),w\rangle \leq \xi_{i},\xi_{i}\geq 0,i=1,\ldots ,n,$$
where *w* is the parameter vector of the separating hyperplane in the kernel feature space, r is the hypersphere radius, and $\xi =\left(\xi_{1},\xi_{2},\ldots ,\xi_{n}\right)$ is the vector of slack variables. The maximal margin $\frac{r}{\left|\left|w\right|\right|}$ is achieved by maximizing the radius and minimizing the weight vector norm. We can summarize the set of OCSVM parameters as the triplet $\theta =\left(w,r,\xi \right),$ and the corresponding hypersphere model $f\left(x;\theta \right)=min\left\{r-\phi_{k}\left(x\right),w,0\right\}$ will then serve as the scoring function for our warning system.

#### 2.3.3. Local Outlier Factor

LOF is an anomaly detection method based on the local density between a data point and its neighboring points. Intuitively, a data point that lies in a densely populated segment of S is likely to be an inlier, whereas a data point that lies in a sparsely populated area is likely to be an outlier. To formalize the notion of density, we first define the reachability distance between two data points as: $$rdk\left(x_{i},x_{j}\right)=max\left\{d\left(x_{j},n\left(x_{j},k\right)\right),d\left(x_{i},x_{j}\right)\right\},$$
where $n\left(x,k\right)$ is the k-th nearest neighbor of a data point x and $d:R^{F}\to R$ is some common distance metric. This definition of reachability distance imposes a lower bound on distances between points, that being the k-th nearest neighbor distance of the second point. This has the sense that a few close points should not matter in quantifying how “reachable” a point is by its adjacent points, thus providing a distance measure that encodes some sense of “global” reachability. Following this, we can define the Local Reachability Density (LRD) at a point of the dataset as the reciprocal mean of its neighbors’ reachability distances:$$lrd\left(x_{i},k\right)=\frac{k}{\sum_{j=1}^{k}rd_{k}\left(x_{i},n\left(x_{i},j\right)\right)},$$
thus, the local reachability density aims to quantify how densely or sparsely populated a neighborhood of the data point is in terms of its neighbors’ reachability. The LOF measure is defined as the average ratio between a point’s k-nearest neighbors’ LRDs and its own LRD:$$LOF\left(x_{i},k\right)=\frac{1}{k}\sum_{j=1}^{k}\frac{lrd\left(x_{i},k\right)}{lrd\left(n\left(x_{i},j\right),k\right)},$$
which quantifies how similar the density of points near a data point is to that of its neighbors. Values less than one correspond to an increase of density near the point in relation to its neighbors, whereas values higher than one correspond to a decrease in local density, indicating outlier or novel behavior; as such, the LOF can be used as a novelty score.

### 2.4. Novelty Scores

A problem with the evaluation of an anomaly detector is that the scoring functions of the investigated models return scores with different interpretations. More specifically, scores for these models cover different domains of real values, so scores that are small for one model may be considered large for another, and vice versa. Even though all the scoring functions we mentioned are positive valued, the sklearn implementations of the models used for this study return negative values for LOF and MCD. This is because these implementations follow a “bigger is better” scoring policy; i.e., large values correspond to inliers. 

To undo the unintuitive nature of the scores calculated by the library models expressing “inlierness”, in figures where these scores are presented, we have “flipped” their values, and min-max scaled so that larger values closer to 1 represent novel samples.

To also examine these models’ suitability as warning systems, we define score thresholds, the crossing of which trigger a warning. For each model, these thresholds need to appear consistent across cases. Due to our observation that MCD and OCSVM score values cover a wider domain than LOF, reporting on their thresholds would yield large deviations across cases, which may be misinterpreted as an inconsistency in performance for each model. A log scaling on the absolute value of their reported scores was used to solve this inconsistency. The negative scores should not be sign inverted by the absolute value operation to avoid altering their meaning. Since MCD is the only model that we apply log scaling to that reports negative scores, we simply correct its sign to be negative post-scaling.

### 2.5. Available Recordings

In this work, we demonstrate through a series of experiments how common anomaly detection algorithms can be applied to track HRV novelty and detect abnormalities that occur prior to partial epileptic seizure onset. These experiments were performed on data from the open dataset “Post-Ictal Heart Rate Oscillations in Partial Epilepsy” (PIHROPE) [50], provided by the Beth Israel Deaconess Medical Center in Boston, Massachusetts, and available on physionet [51]. Originally, this dataset was compiled to investigate the post-ictal alteration of epileptic patients’ heart rates by examining their post-seizure spectrum. The dataset contains single-lead ECG signal recordings of 5 female epileptic patients aged 31–48 sampled at a rate of 200 Hz. Each patient undergoes 1–2 seizures per recording session, making up a total of 10 recorded seizures. As stated in the descriptive document accompanying the dataset, the patients experienced partial seizures with or without secondary generalization from frontal or temporal foci. Annotations that specify the seizure onset and end are provided in the dataset.

### 2.6. Segment Labeling

In evaluating model performance, we need to specify the time intervals in which an ANS disturbance is observable in the patient by labeling the feature series segments as “pre-ictal” or “inter-ictal.” Here, we face some noteworthy challenges. We notice that HRV disturbances set in at a variable time before each patient’s seizure. As such, for each recorded seizure, we have specified a set number of minutes preceding the seizure onset for which all feature series segments are labeled as “pre-ictal.” Segments preceding this pre-ictal interval are labeled “inter-ictal.” In doing so, we set an Optimal Pre-ictal Period (OPP) for each model. A trial into the setting of the OPP for pre-ictal state detection has concluded that detectable alterations in the brain precede the seizure by up to one hour [52], and the cited literature on HRV-based detection has achieved detection of up to 40 min prior to seizure onset. Our designated OPPs for each seizure case were hand-selected based on observation of the NN series, and consideration of the results reported by Perez-Sanchez et al., whose experiments were conducted on the same dataset [53]. Our selected OPPs are reported in Table 2, along with the seizure and reference intervals. It is evident that this labeling procedure may be biased, and, in fact, feature segments that may exhibit characteristics closer to that of an inter-ictal state may be inaccurately labeled as pre-ictal. We also notice that in some seizure cases, patients exhibited a brief disruption of their normal HRV pattern, only to return to a normal condition before suffering the seizure. To tackle these issues, we follow a cluster-then-label approach [52], in which we employ the clustering methodology proposed by Leal et al. [28] to weakly label segments [53] as inter-ictal or pre-ictal. The term “weak labels” is here used to refer to class labels that are assigned to samples by an imperfect procedure and, as such, may contain “noise” (i.e., misclassification error). More specifically, to derive weak labels, we group segments into two clusters and assign segments of one cluster “inter-ictal” weak labels and to the other “pre-ictal” weak labels. In our specific application, we apply a two-phase K-means procedure on the PCA transformed features.

To identify the difference of the assigned classes in terms of sample localization, we calculate their silhouette scores [54] for both hand-picked and weak labels. To investigate the differences between the two label assignment procedures, we calculate their Adjusted Rand Index (ARI) [55]. These metrics are recorded in Table 3. In all cases, a higher silhouette score is reported for K-means assigned weak labels, indicating a more accurate labeling of diverging abnormal samples as opposed to hand-picked labels. For this reason, we postulate that weak labels are more reliable in distinguishing HRV data samples originating from significantly altered cardiac function.

The degree to which hand-picked and weak labels differ is quantified by the ARI, indicating correlated, uncorrelated, or anti-correlated label assignments, by values either 1, 0, and −1, correspondingly. In cases where weak correlation is indicated by the reported ARI, clusters in the feature space differ in terms of the samples that each one includes. In very weakly correlated cases such as sz05-1, this discrepancy between the clusters alters the classification performance significantly, as the set OPP does not reliably reflect the alteration distinguished by the K-means algorithm used in deriving weak labels, and as such, the two sets of labels differ significantly. In other cases where a weak correlation exists, the studied models showed similar performance for hand-picked and weak labels. This may be due to the models’ abilities to handle contamination in the reference segment by a proper setting of hyperparameters.

Finally, the reference intervals were selected by considering the set OPP and pre-ictal labels of each case. In this manner, we facilitate the selection of non-overlapping intervals between pre-ictal patterns and precede its onset by a significant amount of time. As a special case, note that sz01-1 precedes the seizure onset by only 4 min; this might be sub-optimal, but the provided data for this case are too short to do otherwise. We visually inspected the RR series in order to select an interval that is relatively stable, i.e., free from many short fluctuations and disturbances that are seen as sources of contamination. The resulting reference intervals are variable in length, ranging from 4 to 20 min; this, however, does not seem to affect model performance.

### 2.7. Experiment Procedure

Our experiments reflect a realistic scenario in which we first fit the novelty detector on segments extracted from the patient’s NN series during a referential inter-ictal state, effectively calibrating the detector on the patient’s typical HRV pattern; following this, we deploy the detector, feeding it segments of the feature series in a stream until novel HRV behavior is detected and a seizure warning is produced.

For each seizure case, we isolate a 4-to-30-min time interval of feature series segments to use as a reference interval to train the patient-specific anomaly detector model. This interval of time must be contamination-free and representative of normal patient heart rate behavior. In our study, we hand-picked the reference interval for each patient based on prior observation of the NN series. In practical deployment, this interval could be manually annotated by a board-certified epileptologist, specifying an interval reflecting normal ANS function. An example of the novelty score reported by each model using the experimental procedure we describe is presented in Figure 2.

### 2.8. Performance Scores

Another frequent problem in evaluating anomaly detection systems is the choice of threshold for which a certain sample, or in our case a time series segment, is considered an anomaly. In evaluating the novelty score’s relevance in accurately scoring inter-ictal and pre-ictal segments, we employed a threshold-independent evaluation of our system using the area under curve (AUC) metric of the Receiver Operator Characteristic Curve (ROC). Additionally, we report on the performance of an ideal threshold by specifying the *accuracy*, *sensitivity*, and *specificity* measures attained when selecting an optimal threshold in terms of a Balanced Classification Rate (*BCR*) as a fair trade-off between *sensitivity* and *specificity*. Formulae for these measures are as follows:$$Accuracy=\frac{TP+TN}{TP+FP+TN+FN}$$
$$Sensitivity=\frac{TP}{TP+FN}$$
$$Specificity=\frac{TN}{TN+FP}$$
$$BCR=\frac{1}{2}\left(Sensitivity+Specificity\right)$$
where *TP*, *FP*, *TN*, and *FN* denote True Positive, False Positive, True Negative, and False Negative samples, correspondingly. A “positive” sample in this case corresponds to a segment whose novelty score reported by the detector exceeds the specified threshold, whereas a “negative” segment attains a score that remains below this threshold. A “true positive” segment is a positive segment labeled “pre-ictal” whereas a “true negative” is a negative segment labeled “inter-ictal.” For the same threshold, we record the time at which the system crosses this threshold, *triggering* a warning and report on how many minutes/seconds prior to the seizure this warning occurs, which we denote as $\Delta T_{W}$. 

## 3. Results

### 3.1. Experimental Results

The trained models are evaluated using the optimal thresholds on hand-picked and weak labels and their average scores are subsequently recorded. Our results indicate early detection in nine out of ten seizure cases—all but case sz05-1. For this problematic case, a warning is generated 13 min ahead of the seizure; however, it is very likely a false alarm, as the novelty detected results from a very brief disruption in normal behavior, as opposed to all other cases which present a persistent disturbance, and overall, very low performance scores are reported for hand-picked labels in this case. On weak labels, on the other hand, performance is very high, meaning that the K-means algorithm as well as our models registered the brief disturbance as anomalous. This is not incorrect, as a disturbance is present; it is, however, unlikely that this disturbance is epileptic in origin.

To summarize, the models produced meaningful warnings ranging from 3 to nearly 30 min before seizure onset. A per-case breakdown of the performance of the model on each individual can be found in the Appendix A for this paper. Overall, all types of anomaly detectors attained similar scores, with a small advantage for the LOF, which demonstrates a slightly more accurate and consistent warning behavior, as indicated by the average performance scores and their deviation shown in Table 4 (excluding the failed sz05-1 case) as well as the average ROC curve and AUC of the model in Figure 3. Similarly, average ROC curves for the models and their corresponding AUCs are presented in Figure 4, and cumulative confusion matrices are presented in Table 5. 

### 3.2. Post-Ictal Monitoring

The novelty score reported by an anomaly detector allows us to monitor a patient’s heart rate behavior beyond the pre-ictal and ictal periods in order to examine the post-ictal novelty of their HRV parameters. This is useful in detecting persisting autonomic dysfunction post-seizure, as the patient’s HRV parameters may fail to return to a state conforming to the distribution of the reference interval, indicating long-term disruption of cardiac behavior. Patients of the utilized dataset in particular exhibit a distinct Post-Ictal Oscillation (PICO) pattern. Moreover, by investigating the trajectory of the PCA features derived by the series, we notice a certain “seasonality”: the inter-ictal state features are usually localized within a bounded region, which they tend to drift out of when pre-ictal abnormalities occur. The drifting increases during the seizure and, finally, the PCA features gradually drift back within the region, although in cases with a persistent disruption, the features remain outside the region. Examples of these seasonal and disrupted behaviors are presented in Figure 4. More specifically, the NN series of Figure 4a exhibits some seasonal behavior, returning to normal HRV parameters post-seizure, however at a faster rate, as indicated by the slight divergence in the PCA features extracted from the inter-ictal and post-ictal intervals. The NN series of Figure 4b, on the other hand, seems to return to normal HRV parameters post-seizure.

An anomaly detector’s novelty score reflects the course of this measure of HRV “regularity” when the reference interval is set to capture the HRV pattern of the inter-ictal state. More specifically, high novelty scores reflect a deviation from the reference/inter-ictal region in the feature space. On the other hand, to demonstrate the ability of the proposed model to “characterize” different states, we record the average attained scores during inter-ictal, pre-ictal, and post-ictal periods to observe their differences. The min-max scaled average scores attained for the NN series and their derived feature series segments, grouped in accordance with the patient state as specified by the hand-picked labels, are recorded in Table 6 for each experimental case. A more pictorial description of this grouping is provided in the bar plot in Figure 5. Our reason for min-max scaling these scores to fit the [0, 1] range is to obtain a description of the novelty scoring behavior of each model that is consistent across the three models so that they may be examined comparatively using Table 6 and the bar plot in Figure 5.

## 4. Discussion

Overall, all types of anomaly detectors attained similar scores, with a small advantage for the LOF, which demonstrates a slightly more accurate and consistent warning behavior, as indicated by the average performance scores and their deviation shown in Table 4 as well as the average ROC curve and AUC of the model in Figure 3. The average warning time was 14 min; however, this varied widely from case to case. The warning times are included in the Appendix A. The performance was similar for weak and hand-picked labels when an optimal threshold was selected.

An interpretation of the data presented in Table 6 is helpful to understand how these models characterize ictal states by their novelty. We observe a distinct increase in the novelty score during the pre-ictal state, followed by a drop; or the persistence of the score value during the post-ictal state, reflecting a return to a normal state or a persisting disruption. The models tend to differ slightly in their characterization of the different feature trajectory. For example, in the case sz03-1, the LOF and OCSVM exhibit an increase in score during the pre-ictal state. The same, however, does not happen for the MCD, which means that the selected threshold will exhibit suboptimal classification performance, which is consistent with the lower performance scores exhibited for the MCD in this specific case. Another insight gained, by comparison of the barplots for each model, is that the average post-ictal scores for the OCSVM tend to be quite higher, indicating that either this model is highly sensitive to post-seizure HRV alterations or that, in some cases, we overfit the reference interval. To avoid this, we potentially need to apply hyperparameter recalibration, as it is possible that the post-ictal HRV pattern is close enough to the pre-ictal pattern, but the model is too strict in its discrimination. 

In our experiments, we can detect pre-ictal alterations indicative of seizure onset 3 to 30 min prior to onset. The accuracy and timeliness of the seizure warnings produced by the anomaly detectors rivals that reported in similar work using ECG signals, such as that of Popov et al. [56], 20 min/76.2%; Pavei et al. [14], 5 min/95.6%; Billeci et al. [26], 10 min/84.6%; and Giannakakis et al. [25], 21.8 s/77.1%. An added benefit from our proposed pipeline is that it is semi-supervised and immediately deployable without the need of a pre-trained classifier. The 15 min warning prior to seizure time reported in the work of Perez et al. on the same dataset is close to the performance attained by our system, which supports our findings. In [57], the authors extracted features using Wavelet Transform and reported 100% accuracy, being able to predict seizures in all cases. However, the training set used in this study included segments of the pre-ictal HRV pattern for each case that the model was tested on. Encoding prior knowledge of the pre-ictal HRV pattern of a specific case into the model by including it in the training set is likely to produce overfitted results, exhibiting high accuracy on the training set that proper cross-validation would not corroborate. In our experiments, the training sets consist of only a small sample of each patient NN series that does not contain pre-ictal segments, encoding no prior knowledge of this pattern in the model. The performance of our models meets the required standards set by patients and health staff when surveyed regarding the capabilities of such a system [57] with a 3–5 min warning., Warnings of up to 25 min seem to satisfy the majority of patients and caregivers. We also demonstrated how a novelty measure can be used to monitor patient cardiac function and identify persistent abnormalities post-seizure that may result in an overall decline in patient health. This is an additional feature of our proposed framework, with potential use in clinical or personalized medicine applications, which the mentioned publications do not include.

It is important to clarify that our findings indicate a pre-ictal increase in HRV novelty that is linked to ANS abnormalities; this is not, however, the only possible source of novelty in an NNI series. As has already been discussed by Giannakakis and colleagues [25], heart rate is a highly variable modality, altering in relation to physical activity, psychological condition, etc., which may induce false alarms if not provided as part of the reference interval. Specifically, in [27], an experimental case is described in which a patient’s heart rate was significantly disrupted by the patient performing a trivial activity, greatly altering system performance. Our novelty-sensitive design may be fundamentally compromised in such scenarios, possibly limiting its reliability. Since HRV parameters are known to fluctuate throughout the day, corresponding to different bodily activities, it would be highly beneficial for a study on longer HRV data to examine the effect of these fluctuations on novelty-based seizure prediction. We suggest that the models may need to be trained on more than one reference interval to avoid misclassification. Our study is, however, limited by the short length of the examined dataset in this regard.

We clarify that our models have not been tested on body sensor data of active patients, so it is unknown how motion artifacts as well as other sensor noise sources, linked with ECG and estimated RR interval unreliability, will affect their performance. An analysis of the effect is beyond the scope of this study; however, we cite two studies indicating some usability of noisy ECG data in extracting HRV [58,59]. We propose that recent methods developed for motion artifact removal and robust R peak estimation [60,61,62,63,64] be used in conjunction with the proposed anomaly detectors, if practical deployment is to be carried out.

We also note that our dataset includes only samples of five female subjects aged 31 to 48 years [50]; however, HRV parameters are known to differ across the age spectrum and biological sex [65]. For epileptic short-term HRV, we indicatively cite two studies that found differentiations in inter-ictal heart rate patterns across age and gender [66], as well as gender differences in pre-ictal heart rate patterns [67]. Another notable limitation of our work stemming from the size of the dataset is the availability of recordings and related metadata from diverse seizure types, which would allow us to conduct a subgroup analysis of the classification performance of our models. Our intuition, based on the models’ function, is that the patient-specific deployment we propose will successfully identify occurring HRV novelties, provided that a stable enough reference interval is set, and the abnormality is quantifiably significant, regardless of the range of HRV characteristics contained in this reference interval. We cannot, however, formally justify this currently due to the limited size of our dataset. It could be that certain seizure-related HRV alterations do not present a novel enough pattern for our models to discriminate. Combined, these concerns prompt the need for further validation of our methodology in a wider demographic of epileptic patients. 

In our approach, we also make use of a novelty score that requires the selection of optimal thresholds to produce warnings. This may not be concise across different patient cases. Each one of the models we examined exhibited some consistency regarding the selection and its range of appropriate thresholds; these threshold, however, may need to be fine-tuned to attain optimal results. This could necessitate precise setting by the patient or an expert and may also require recalibration, all being quite impractical. Similar effects in the performance of the system with respect to the provided training set are observed in [27]. The development of a fully supervised approach that does not require setting a reference interval and is consistent across multiple seizure cases could dispose of these concerns. In the present models, the use of a novelty score could serve as a useful indicator of the cardiac stability of epileptic patients, should doctors or patients deem that specifying a threshold proves inconvenient and or unreliable.

Finally, it may be that the pursuit of predicting seizure onset using ECG data alone is a fool’s errand. However promising recent approaches in this domain may appear, it is still not known how these models, including the ones we propose, fare against other types of abnormal heart rate disruptions that are unrelated to epilepsy. The incorporation of other signal modalities, such as EEG [68] accelerometry, electrodermal activity, electromyography, photoplethysmography, etc. [69], may be vital for robust characterization of a patient state as definitively pre-ictal.

## 5. Conclusions

In conclusion, we indicate how a set of approaches from the current state-of-the-art of ECG/HRV-based seizure detection, particularly those utilizing machine learning, may suffer from certain design flaws, limiting their potential to be applied in practice. We propose that these problems can be treated by applying semi-supervised or unsupervised approaches of the same ilk, and demonstrate how certain anomaly detection methods, namely LOF, MCD, and OCSVM, can be used in this fashion. The experimental results yielded by these models should prompt future inquiry regarding the use of similar methods in the context of epileptic seizure prediction and monitoring. Most studies thus far, including ours, are limited to a small number of patient cases and short ECG recordings, which limits their scope. A more systematic investigation necessitates larger datasets of quality ECG recordings from a wide variety of patient cases. Proposed research directions to be taken into consideration going forward in improving the state of the art are the extraction of robust features, the deployment of similar, potentially more powerful models for detection, investigation into secondary biosignal modalities to incorporate into the abnormality detection pipeline, as well as research into possible completely unsupervised techniques for detecting novel pattens associated with epilepsy. 

## Figures and Tables

**Figure 1 ijerph-20-05000-f001:**
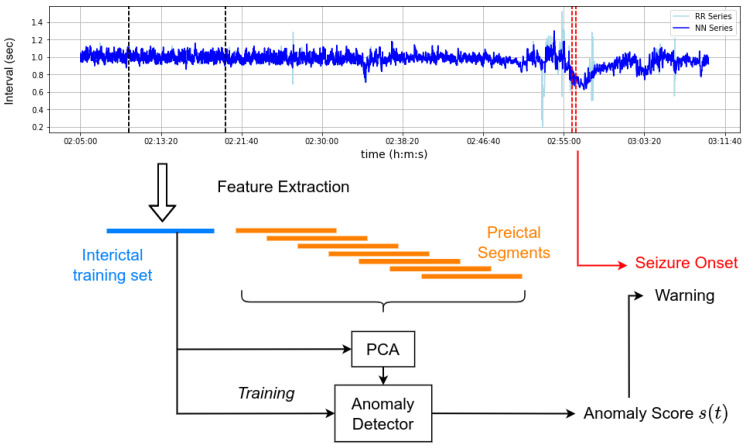
NN series novelty monitoring and ahead-of-time prediction scheme.

**Figure 2 ijerph-20-05000-f002:**
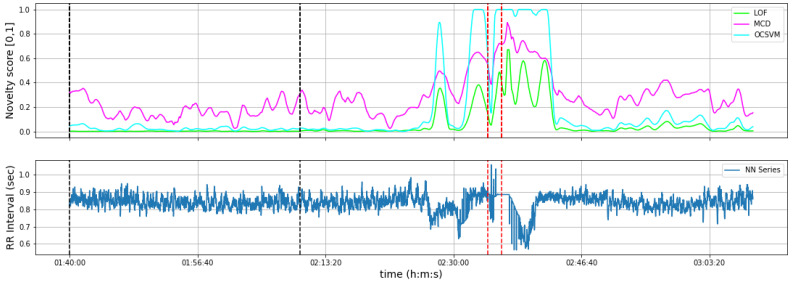
NN series novelty monitoring examples. Anomaly scores are averaged over 50 consecutive segments and min-max scaled. The black dashed lines correspond to the reference interval whereas the red lines correspond to the seizure event.

**Figure 3 ijerph-20-05000-f003:**
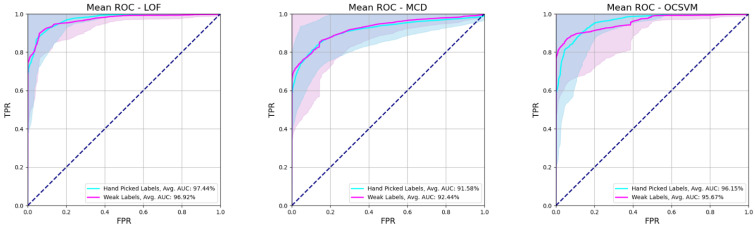
Average ROC curves and AUC for the three models across all experimental cases. Shaded areas reflect the standard deviation of the curves.

**Figure 4 ijerph-20-05000-f004:**
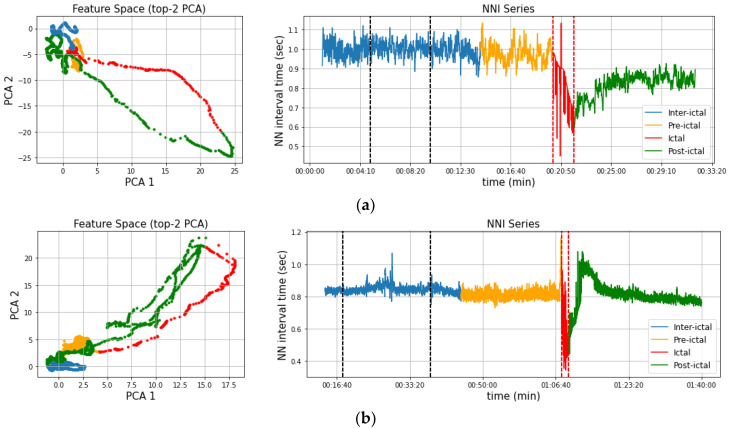
Examples of NN series and the corresponding top-2 PCA features that were extracted from them; seasonality and subfigures correspond to cases sz04-1 (**a**) and sz07-1 (**b**). The black dashed lines correspond to the reference interval whereas the red lines correspond to the seizure event.

**Figure 5 ijerph-20-05000-f005:**
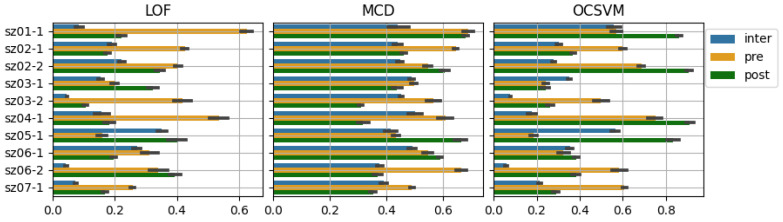
Bar plot of min-max scaled mean score values per detector for different patient states; deviation is depicted with gray bars.

**Table 1 ijerph-20-05000-t001:** HRV Statistical Features.

Feature	Description	Formula ^1^	Unit ^2^
MEAN	Mean of NN intervals	$\frac{1}{n}\sum_{j=1}^{n}NN_{j\;}$	$\left[\mathrm{ms}\right]$
SD	Standard deviation	$\sqrt{\frac{1}{n-1}\sum_{j=1}^{n}{\left(NN_{j}-\bar{NN_{j}}\right)}^{2}}$	$\left[\mathrm{ms}\right]$
SKEW	Skewness of NN intervals	$\frac{1}{n-1}\sum_{j=1}^{n}{\left(\frac{NN_{j}-\bar{NN_{j}}}{SD}\right)}^{3}$	
KURT	Kurtosis of NN intervals	$\frac{1}{n-1}\sum_{j=1}^{n}{\left(\frac{NN_{j}-\bar{NN_{j}}}{SD}\right)}^{4}\;$	
NNX	Number of ΔNN exceeding the threshold X	$\left(\Delta {NN}_{j}<X\right)$	
SDSD	Standard deviation of ΔNN	$\sqrt{\frac{1}{n-1}\sum_{j=1}^{n}{\left(\Delta NN_{j}-\bar{\Delta NN_{j}}\right)}^{2}}$	$\left[\mathrm{ms}\right]$
RMSSD	Root mean of squared differences of NNI	$\sqrt{\frac{1}{n-1}\sum_{j=1}^{n}\Delta NN_{j}^{2}}$	$\left[\mathrm{ms}\right]$
SAMPEN	Sample entropy [40]	(See cited paper)	
SD1	Standard deviation of major axis in Poincaré plot [41]	$\sqrt{\frac{1}{2}SDSD^{2}}$	$\left[\mathrm{ms}\right]$
SD2	Standard deviation of minor axis in Poincare plot	$\sqrt{2SDNN^{2}-\frac{1}{2}SDSD^{2}}$	$\left[\mathrm{ms}\right]$
SD1SD2	Ratio between SD1, SD2	$\frac{SD2}{SD1}$	
ELLIPSE	Ellipse area of Poincare plot	π⋅SD1⋅SD2	$\left[{\mathrm{ms}}^{2}\right]$
KFD	Katz fractal dimension [42]	$log\left(n\right)\cdot \left(log\left(n\right)+log\left(\frac{d}{L}\right)\right)$	
LF	Low-frequency band energy (0.04–0.15 Hz)	$\int_{LFB}PSD_{NN}^{LFB}\left(f\right)df$	[${\mathrm{ms}}^{2}$]
HF	High-frequency band energy (0.15–0.4 Hz)	$\int_{HFB}PSD_{NN}^{HFB}\left(f\right)df$	$\left[{\mathrm{ms}}^{2}\right]$
LFHF	Low-frequency to high-frequency band energy ratio	$\frac{LF}{HF}$	
LFPEAK	Peak LF band frequency	${\mathrm{argmax}}_{f}\left\{PSD_{NN}^{LFB}\left(f\right)\right\}$	$\left[\mathrm{Hz}\right]$
HFPEAK	Peak HF band frequency	${\mathrm{argmax}}_{f}\left\{PSD_{NN}^{HFB}\left(f\right)\right\}$	$\left[\mathrm{Hz}\right]$

^1^ Here, n denotes NN series segment length. ^2^ Unit abbreviations: ms = milliseconds, Hz = Hertz.

**Table 2 ijerph-20-05000-t002:** Recording seizure ictal, reference intervals and selected OPP for each case.

Case	Seizure ^1^		Reference ^1^		OPP ^2^
	Start	End	Start	End	
sz01-1	00:14:36	00:16:12	00:04:00	00:08:00	6:30
sz02-1	01:02:43	01:03:43	00:15:00	00:25:00	30:00
sz02-2	02:55:51	02:56:16	02:10:00	02:20:00	21:00
sz03-1	01:24:34	01:26:22	00:30:00	00:50:00	22:00
sz03-2	02:34:27	02:36:17	01:40:00	02:10:00	7:20
sz04-1	00:20:10	00:21:55	00:05:00	00:10:00	6:00
sz05-1	00:24:07	00:25:30	00:03:30	00:10:00	13:30
sz06-1	00:51:2	00:52:19	00:30:00	00:35:00	7:00
sz06-2	02:04:45	02:06:10	01:43:00	01:50:00	6:30
sz07-1	01:08:02	01:09:31	00:18:00	00:38:00	23:00

^1^ Label format is hours:minutes:seconds. ^2^ OPP format is minutes: seconds.

**Table 3 ijerph-20-05000-t003:** Silhouette and ARI scores.

Case	Silhouette		ARI
	Weak	Hand-Picked	
sz01-1	0.533	0.503	0.741
sz02-1	0.414	0.356	0.588
sz02-2	0.400	0.357	0.674
sz03-1	0.367	0.313	0.758
sz03-2	0.758	0.604	0.644
sz04-1	0.552	0.498	0.664
sz05-1	0.631	0.005	−0.054
sz06-1	0.418	0.247	0.131
sz06-2	0.782	0.475	0.344
sz07-1	0.723	0.714	0.970

**Table 4 ijerph-20-05000-t004:** Average model performance (mean ± std) across all PIHROPE ^1^.

Model	Case	Threshold	$\Delta T_{W}\;{\;}^{2}$	AUC (%)	Accuracy (%)	Sensitivity (%)	Specificity (%)
*LOF*	*Hand-picked*	−1.77 ± 0.64	14:07 ± 8:40	97.4 ± 3.1	94.8 ± 4.8	93.0 ± 7.7	95.8 ± 5.3
	*Weak*	−2.53 ± 1.17	14:01 ± 8:32	96.9 ± 3.8	93.5 ± 7.7	95.6 ± 5.2	93.1 ± 9.3
*MCD*	*Hand-picked*	−2.07 ± 0.54	14:06 ± 8:41	91.6 ± 7.9	86.9 ± 9.4	88.2 ± 8.3	85.7 ± 14.7
	*Weak*	−2.20 ± 0.53	13:37 ± 8:02	92.4 ± 8.9	88.9 ± 9.3	91.1 ± 9.2	87.8 ± 10.5
*OCSVM*	*Hand-picked*	1.85 ± 0.50	14:05 ± 8:38	96.1 ± 5.5	93.2 ± 6.5	92.7 ± 7.9	93.4 ± 7.5
	*Weak*	1.47 ± 0.63	14:00 ± 8:35	95.6 ± 7.9	95.6 ± 4.2	92.4 ± 15.7	96.6 ± 4.6

^1^ Metrics are referenced in the format: hand-picked label metrics/weak label metrics. ^2^ Time format: minutes: seconds.

**Table 5 ijerph-20-05000-t005:** Cumulative percentile confusion matrices for the three models across all experimental cases evaluated with (**a**) hand-picked and (**b**) weak labels.

**(a) Hand-Picked Labels**								
**LOF (%)**	**Pre-Ictal**	**Inter-Ictal**	***MCD* (%)**	**Pre-Ictal**	**Inter-Ictal**	***OCSVM* (%)**	**Pre-Ictal**	**Inter-Ictal**
+	36.60	2	+	34.30	8	+	36.48	4
-	5.47	56	-	7.74	50	-	5.85	54
**(b) Weak Labels**								
**LOF (%)**	**Pre-Ictal**	**Inter-Ictal**	***MCD* (%)**	**Pre-Ictal**	**Inter-Ictal**	***OCSVM* (%)**	**Pre-Ictal**	**Inter-Ictal**
+	33.78	4	+	31.87	9	+	34.91	2
-	1.60	60	-	3.69	56	-	1.81	61

**Table 6 ijerph-20-05000-t006:** Min-max scaled average score (± standard deviation) at different ictal states per model ^1^.

Case	State	*LOF*	*MCD*	*OCSVM*
	*Inter-ictal*	0.09 ± 0.12	0.19 ± 0.28	0.57 ± 0.27
sz01-1	*Post-ictal*	0.22 ± 0.24	0.17 ± 0.21	0.92 ± 0.19
	*Pre-ictal*	0.63 ± 0.22	0.41 ± 0.29	0.69 ± 0.25
	*Inter-ictal*	0.19 ± 0.18	0.20 ± 0.24	0.36 ± 0.20
sz02-1	*Post-ictal*	0.18 ± 0.19	0.15 ± 0.18	0.70 ± 0.26
	*Pre-ictal*	0.43 ± 0.24	0.28 ± 0.19	0.86 ± 0.22
	*Inter-ictal*	0.22 ± 0.20	0.15 ± 0.18	0.36 ± 0.18
sz02-2	*Post-ictal*	0.35 ± 0.21	0.30 ± 0.23	0.93 ± 0.19
	*Pre-ictal*	0.40 ± 0.22	0.28 ± 0.23	0.87 ± 0.15
	*Inter-ictal*	0.15 ± 0.18	0.19 ± 0.20	0.40 ± 0.22
sz03-1	*Post-ictal*	0.32 ± 0.25	0.19 ± 0.24	0.51 ± 0.27
	*Pre-ictal*	0.20 ± 0.20	0.14 ± 0.19	0.60 ± 0.23
	*Inter-ictal*	0.05 ± 0.06	0.16 ± 0.16	0.16 ± 0.12
sz03-2	*Post-ictal*	0.11 ± 0.18	0.02 ± 0.09	0.47 ± 0.32
	*Pre-ictal*	0.42 ± 0.35	0.30 ± 0.31	0.68 ± 0.40
	*Inter-ictal*	0.16 ± 0.28	0.16 ± 0.26	0.32 ± 0.29
sz04-1	*Post-ictal*	0.18 ± 0.25	0.17 ± 0.22	0.92 ± 0.22
	*Pre-ictal*	0.53 ± 0.28	0.41 ± 0.27	0.89 ± 0.20
	*Inter-ictal*	0.35 ± 0.21	0.13 ± 0.21	0.60 ± 0.22
sz05-1	*Post-ictal*	0.40 ± 0.34	0.34 ± 0.33	0.91 ± 0.19
	*Pre-ictal*	0.16 ± 0.28	0.09 ± 0.15	0.29 ± 0.29
	*Inter-ictal*	0.27 ± 0.21	0.13 ± 0.15	0.41 ± 0.21
sz06-1	*Post-ictal*	0.20 ± 0.23	0.16 ± 0.19	0.70 ± 0.26
	*Pre-ictal*	0.31 ± 0.28	0.34 ± 0.18	0.51 ± 0.30
	*Inter-ictal*	0.04 ± 0.09	0.04 ± 0.08	0.21 ± 0.14
sz06-2	*Post-ictal*	0.39 ± 0.25	0.10 ± 0.14	0.68 ± 0.20
	*Pre-ictal*	0.34 ± 0.32	0.34 ± 0.22	0.80 ± 0.24
	*Inter-ictal*	0.07 ± 0.10	0.10 ± 0.21	0.27 ± 0.20
sz07-1	*Post-ictal*	0.17 ± 0.19	0.08 ± 0.20	0.54 ± 0.32
	*Pre-ictal*	0.26 ± 0.15	0.15 ± 0.11	0.94 ± 0.10

^1^ Average score and standard deviations are min-max scaled to fit the [0, 1] range.

## Data Availability

Data available in a publicly accessible repository. The data presented in this study are openly available on physionet at https://doi.org/10.13026/C2QC72 (accessed on 2 December 2022).

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
