# Peer review of "ECG-Based Semi-Supervised Anomaly Detection for Early Detection and Monitoring of Epileptic Seizures"

_ijerph, 2023, doi:10.3390/ijerph20065000_

Round 1

Reviewer 1 Report

This paper proposed an epileptic seizure prediction method based on the anomaly detection of heart rate variability (HRV) markers. As hand-picked method was applied, the authors also introduced optimal pre-ictal period (OPP) and clustering methodology for data labeling. Then they applied principal component analysis (PCA) to derive HRV features used for anomaly detection, where the anomaly scores were then used for seizure warnings. The experiment results indicated that the proposed anomaly detection and monitoring approach could enable early detection and warning of seizure. The authors provide a detailed method description. Still, some concerns we have are listed as the following:

1. In page 2, the authors mentioned “when designing a patient-specific system a patient would have to enroll by undergoing a long ECG recording session” as a major concern of prior studies. In fact, ECG signals as well as HRV could have characteristic fluctuations throughout the day. As the authors indicated the use of a small sample set was an advantage of the proposed method, have you investigated the influence of data length to the novelty quantification of HRV unlike other studies. Please justify.

2. In page 2, since HRV has a clear medical definition that should derived by long-term ECG signals, the corresponding term used in the third paragraph should be short-term HRV as many studies have demonstrated the difference between the two. Calculating of HRV features using a window that is too short would make HRV statistical features lose their medical meanings, while the authors desire to provide a medical interpretation of the method through the link between HRV and autonomic nervous system. Please justify.
[1] Nunan, D., Sandercock, G. R., & Brodie, D. A. (2010). A quantitative systematic review of normal values for short‐term heart rate variability in healthy adults. Pacing and clinical electrophysiology, 33(11), 1407-1417.
[2] Sandercock, G. R., Bromley, P. D., & Brodie, D. A. (2005). The reliability of short-term measurements of heart rate variability. International journal of cardiology, 103(3), 238-247.
[3] Nunan, D., Donovan, G. A. Y., Jakovljevic, D. G., Hodges, L. D., Sandercock, G. R., & Brodie, D. A. (2009). Validity and reliability of short-term heart-rate variability from the Polar S810. Medicine & Science in Sports & Exercise, 41(1), 243-250.

3. In page 3, the authors applied minimal pre-processing to the raw ECG signal as they only concerned the R-peak detection results. However, when talking about real world mobile ECG data collected by wearable devices mentioned in the Introduction Section, noises such as motion and electromyography artifacts under daily monitoring conditions will greatly affect the accuracy of R-peak detection, resulting in ineffective calculation of HRV parameters. Therefore, the proposed pre-processing method was insignificant for realistic scenario. Since pre-processing is not the main research purpose of this study, we would like to see an analysis of the impact of R-peak detection accuracy on the assessment of HRV novelty.

4. In page 4, the authors should distinguish the concept of RR interval from NN interval more clearly and use both terms accurately in the text, where RR intervals refer to interbeat intervals between all successive heartbeats, while NN intervals refer to interbeat intervals from which artifacts have been removed. The title or legend in Figure 1 also needs to be modified.

5. In page 9, the adjusted rand indices indicated weak correlation between hand-picked and weak labels. Does this have an additional impact on model training? Whether there is a significant difference between the two in the feature space? And how to explain the similar classification performance with the difference between the two?

6. In page 12, the “Average Score” in Table 2 was confusing since this table was not mentioned in the text and the sequence number was wrong. Please justify.

7. In page 14, the author mentioned “It is important however to note that these results stem from experiments in which the entirety of the dataset was used as training data, whereas in our experiments our training sets consist of only a small sample of each patient NN series.” Would the use of the entirety of the dataset help improve algorithm performance?

8. Due to the limited amount of database data, the method has only been validated on small datasets. However, this limitation should be mentioned more prominently. Or the author could provide more data support to prove the effectiveness of the method, such as more examples of patient results in the feature space or normalized results, rather than the results of two certain individuals in Figure 4.

Reviewer 2 Report

This paper describes a new approach for early detection and monitoring of epileptic seizures, using ECG instead of EEG and unary (novelty detection) instead of binary classifiers. Both of the two ingredients have been studied before in relation to epileptic seizures, but the combination seems to be new.

1. The dataset is limited to 10 seizure events collected from 7 patients. For each seizure event, the author set an optimal pre-ictal period for each model with a unique reference interval. It is not clear how these reference intervals were selected or what criteria were used to determine their length and timing. The selection of reference intervals is crucial and should be made clearer.

2. The anomaly detection algorithm requires a specified threshold to determine "positive" and "negative" cases. What was the threshold, and how was it established? Was the same threshold used for all models or did each model have a unique threshold? More explanation is needed in the method section.

3. The algorithm uses ECG for early detection of seizure events, but ECG signals, especially HRV features derived from ECG signals, could vary due to patients' cardiac conditions. For example, cardiac arrhythmias can cause convulsive syncope, which is a seizure-like episode. It would be interesting to see how well the algorithm can differentiate between these events.

4. For the formula for the HRV features, it would be much clearer to change the notation for the number of intervals from uppercase N to lowercase n, to avoid confusion with NN intervals, which also involves uppercase N.

Reviewer 3 Report

The authors introduced a study on using anomaly detection models to predict seizures in epilepsy patients. The researchers used pre-ictal changes in heart rate variability from ECG signals to classify a patient's condition using machine learning models such as One-Class SVM, Minimum Covariance Determinant Estimator, and Local Outlier Factor. The models were evaluated on the PIHROPE dataset and achieved detection in 9 out of 10 cases with high average AUCs and warning times ranging from 6 to 30 minutes before the seizure. The proposed approach uses low supervision requirements and has the potential to lead to early detection and warning of seizures based on other body sensor inputs.

1. The first line of the introduction section, “In 2022, there were over 50 million sufferers of Epilepsy worldwide”, the first letter of Epilepsy should be lowercase.

2. “Unsupervised identification of the pre-ictal interval using HRV has been carried out by

Leal et al. [31] where they examined different clustering algorithms to identify the preictal

interval in recorded seizures. However, their proposed framework is not developed

to be used for real-time detection.” The authors need to specify why this framework cannot be used for real-time detection.

3. The dataset contains ECG signal recordings of seven female epileptic patients, ages sampled at a rate of 200 Hz, each undergoing 1-2 seizures per recording session making up a total of ten recorded seizures. The dataset only includes female subjects. Is this the study's limitation? The introduction of dataset should be improved.

4. 3.1. Experiment procedure and 3.2. Performance Scores should be part of the methods.

5. Discussion and Conclusion should be discussed separately.

6. AUC-ROC curve should be reported and the optimal threshold should be reported too.

7. The confusion matrix and misclassification cases should be discussed too.

8. A native English speaker should correct the programmer, typos, and other manuscript writing problems.

Round 2

Reviewer 1 Report

I am satisfied how you fulfilled previous requirements. I am suggesting manuscript acceptance.

Reviewer 3 Report

The issues have been addressed.